# Exposure to the 1959–1961 Chinese famine and risk of non-communicable diseases in later life: A life course perspective

**Mengling Cheng[1]\*, Nicolas Sommet[1], Marko Kerac**  **[2,3], Daniela S. Jopp[1,4], Dario Spini[1,4]**

**1** Swiss Centre of Expertise in Life Course Research, Faculty of Social and Political Sciences, University of Lausanne, Lausanne, Switzerland, **2** Centre for Maternal Adolescent Reproductive & Child Health (MARCH), London School of Hygiene & Tropical Medicine, London, United Kingdom, **3** Department of Population Health, London School of Hygiene & Tropical Medicine, London, United Kingdom, **4** Institute of Psychology, University of Lausanne, Lausanne, Switzerland

\* mengling.cheng@unil.ch

## Abstract

Child undernutrition and later-life non-communicable diseases (NCDs) are major global health issues. Literature suggests that undernutrition/famine exposure in childhood has immediate and long-term adverse health consequences. However, many studies have theoretical and methodological limitations. To add to the literature and overcome some of these limitations, we adopted a life course perspective and used more robust methods. We investigated the association between exposure to the 1959–1961 Chinese famine and later-life NCDs and if this association depends on: life stage at exposure, famine severity, and sex. We conducted a secondary data analysis of a large-scale, nationally representative, longitudinal study—the China Health and Retirement Longitudinal Study (2011–2018, 11,094 participants). We measured famine exposure/severity using self-reported experience, life stage using age at exposure, and health using the number of NCDs. We performed Poisson growth curve models. We obtained three findings. First, compared with unexposed participants, those exposed before age 18 had a higher risk of later-life NCDs, particularly if exposed in-utero (IRR = 1.90, 95% CI [1.70, 2.12], $p < .001$) and in the "first 1,000 days" of life (IRR = 1.86, 95% CI [1.73, 2.00], $p < .001$; for 0–6 months group, IRR = 1.95, 95% CI [1.67, 2.29], $p < .001$). Second, the famine effects among participants moderately and severely exposed were similar (IRR = 1.18, 95% CI [1.09, 1.28], $p < .001$ and IRR = 1.24, 95% CI [1.17, 1.32], $p < .001$). Third, the famine effects did not differ between females and males (IRR = 0.98, 95% CI [0.90, 1.07], $p = .703$). In an individual's life course, in-utero and the "first 1,000 days" are a particularly sensitive time period with marked long-term implications for NCDs if undernutrition/famine is experienced in this period. However, this window remains open until young adulthood. This highlights the need to invest more in preventing and treating child/adolescent undernutrition to tackle later-life NCDs.

**Data Availability Statement:** Data derived from a source in the public domain. For more information on the CHARLS dataset, please refer to http://charls.pku.edu.cn/en/Data/Harmonized_CHARLS.

htm. The R scripts to reproduce our findings are available via the Open Science Framework (OSF): https://osf.io/6zy43/.

**Funding:** Mengling Cheng acknowledges funding from the Swiss National Centre of Competence in Research "LIVES - Overcoming vulnerability: Life course perspectives" financed by the Swiss National Science Foundation (51NF40-185901) and the European Union Horizon 2020 Research and Innovation Programme under the Marie Skłodowska-Curie Grant (801076). Marko Kerac also gratefully acknowledges UKRI GCRF / Medical Research Council funding (grant reference MR/V000802/1). The funders had no role in study design, data collection and analysis, decision to publish, or preparation of the manuscript.

**Competing interests:** The authors have declared that no competing interests exist.

## Introduction

Globally, 194 million children aged under five years suffer from undernutrition and undernutrition accounts for around 45% of deaths in the age group [1, 2]. Conflict, climate crisis and COVID-19 further exacerbate the strain on undernutrition prevention and treatment services [3]. Non-communicable diseases (NCDs) are another major global issue accounting for 71% of all deaths [4].

Existing literature suggests that undernutrition in childhood has immediate detrimental and long-term adverse health consequences [5], including increasing the risk of NCDs in later life [6]. Such long-term sequelae can be explained by both biological and socioeconomic pathways. Biological explanations include the "Developmental Origins of Health and Disease" hypothesis [7, 8] and the "Capacity-load" model [9]: individuals who experienced undernutrition in early years of life have impaired metabolic and organ capacity, which increases the risk of NCDs in later life. Socioeconomic explanations include life-long consequences of early-life poverty: a greater risk of undernutrition in childhood leads to impaired development, suboptimal educational achievement and earning, and loss of full human potential [10, 11]. These consequences of early-life poverty are associated with poverty in adulthood, which independently increases the risk of NCDs in later life [12].

One body of the literature has explored the long-term consequences of famine exposure, and how that links to NCDs in later life. This includes studies of the Dutch Hunger Winter (1944–1945), the Siege of Leningrad (1941–1944), the Chinese Famine (1959–1961), and famines in other countries [6, 13, 14]. However, many existing studies have theoretical and methodological limitations.

### Theoretical limitations: Lack of a life course perspective

From a theoretical angle, many existing studies on the association between famine exposure and later-life health lack a life course perspective. First, they often focus on individuals exposed to famine during a narrow period of life, often on the "first 1,000 days" of life [15]. This is problematic because other evidence suggests that the window of development and growth may extend beyond these "first 1,000 days" [16, 17]. Second, other studies do not always consider the age at which individuals were exposed to famine [13]. Life course epidemiology suggests that the timing of exposure is critical and that the effect of exposure during a specific period may differ from the effect of the same exposure during another period [18, 19]. Thus, individuals exposed to famine at different life stages may be affected differently.

### Methodological limitations: Information bias and survivor bias

From a methodological angle, the observed association between famine exposure and later-life health in many existing studies may be subject to information bias and survivor bias. A major source of information bias is the misclassification of famine exposure and/or severity of famine exposure [6]. To determine famine exposure, with few exceptions [20], many studies classify participants by their birth year: participants born during the famine are classified as exposed, and those born after as unexposed. However, given that older age is a risk factor for NCDs [21], a consequence is that the uncontrolled age difference between participants born during the famine (i.e., older) and participants born after the famine (i.e., younger) potentially biases the health effects attributed to famine exposure. This issue is particularly severe for studies on famines that lasted for several years (i.e., entailing a nontrivial age difference between participants born during the famine and born after the famine), such as the Chinese famine of 1959–1961.

To determine severity of famine exposure, with few exceptions [22], many studies classify participants by regional mortality: participants born in regions where excess mortality during famine was above a predefined threshold are classified as severely exposed, and those born in regions where excess mortality was below that threshold are classified as moderately exposed. Using this approach may lead to misclassification issues because there might be important within-region variations. For example, the food rationing system preferentially supplied urban residents at the time of the Chinese famine of 1959–1961; because they were not entitled to additional food through the rationing system, rural residents within the same region were therefore affected more by the famine than urban residents [23]. This type of information bias highlights the need to avoid classification based on year of birth or region.

In addition, another potential source of bias is survivor bias among famine survivors [13]. A recent review suggests that boys are biologically more vulnerable to undernutrition than girls [24]. However, studies on the Chinese famine of 1959–1961 suggest that due to a culture of son preference, families may have preferentially allocated food to sons over daughters [25]. The implication is that the males exposed to famine might be less affected and thus more likely to survive than their female counterparts. In the context of the Chinese famine of 1959–1961, this type of survivor bias highlights the need to explore the possible moderating role of sex.

## Overview of the study

The current study of a large-scale, nationally representative, longitudinal study ($\approx$ 26,000 participants) aimed to add to the literature and overcome some of the limitations found in previous research, that is, the lack of a life course perspective and information bias. Specifically, we adopted a life course perspective by focusing on a wide period of life ranging from the in-utero to adulthood (24–40 years) and considering the age at famine exposure. Moreover, we used more robust methods to adjust for age/cohort and used a more direct measure of famine exposure/severity. Our overall aim was to understand the link between exposure to the Chinese famine of 1959–1961 and NCDs 50 years later in 2011–2018. Our objectives were to determine if the link between famine exposure and NCDs in later life varied among individuals exposed at different life stages, individuals exposed to different levels of famine severity, and women and men.

## Methods

We conducted a secondary analysis of a large-scale, nationally representative, longitudinal study, which provides both prospective data and retrospective data: the China Health and Retirement Longitudinal Study (CHARLS), which followed Chinese urban and rural residents aged 45 years old and above from 2011 to 2018. We measured famine exposure and severity of famine exposure using the individual self-reported famine experience, life stages using the age at famine exposure, and later-life health using the number of NCDs. We built a series of Poisson growth curve models to investigate the association between famine exposure and later-life NCDs. Specifically, to understand if this association depends on the life stage (age) at which individuals were exposed, we used two analytical approaches that have been used in the few existing studies on the topic: the factorial approach (i.e., the interaction between famine exposure and life stages) and a more parsimonious approach—the concatenation approach (i.e., the famine exposure-life stages cohort). We also conducted two sets of sensitivity analysis to test the robustness of the association between famine exposure, life stage, and later-life NCDs. In addition, we tested the role of severity of famine exposure and sex.

## Study setting: The Chinese famine of 1959–1961

More than half a century ago, China experienced the largest famine in human history, in terms of duration, nationwide geographic scope, and the number of individuals affected [26, 27].

The famine lasted for three years from the spring of 1959 to the winter of 1961 throughout the country. During the famine years, grain production dropped dramatically by about 25% and nearly 30 million people died prematurely from famine [28]. Grain production and grain production per capita returned to the pre-famine level in 1965 and after 1970, respectively [26], indicating a slow and steady recovery to normal food intake.

## Data and participants

We used the data from the CHARLS, a large-scale, nationally representative, longitudinal study that provides both prospective data and retrospective data. CHARLS collects prospective health data from Chinese urban and rural residents aged 45 years old and above biennially since 2011. CHARLS applies a stratified, multistage probability sampling design [29]. First, CHARLS uses a probability proportion to size (PPS) sampling approach for all county-level unit except for Tibet after stratification by region, county characteristic (urban or rural), and per-capita gross domestic product. Primary sampling units (PSUs) are administrative villages in rural areas and neighborhoods in urban areas. Second, CHARLS uses CHARLS-GIS software to randomly sample households within each PSU. Last, within each sampled household, if occupants were aged above 40, one of them is randomly selected. If the selected person was aged 45 or above, they become the main respondent and their spouse is also interviewed; if the selected person was aged between 40 and 44, they are reserved as a refreshment sample. If an age-eligible person was too frail to answer questions, a proxy respondent is identified to help them to answer questions. The sample is representative of participants aged 45 and above living in households. In our analyses, we merged the retrospective Life History data collected in 2014 with all four regular data waves collected in 2011, 2013, 2015, and 2018 (25,863 participants).

We included eligible participants based on two a-priori criteria: (i) participants with complete information on NCDs, famine exposure, sociodemographic variables, childhood family financial status, and childhood and adulthood health status; and (ii) participants born between 1 January 1919 and 30 September 1960 (i.e., in-utero to age of 40 at famine exposure).

## Ethics

This study conformed to the principles laid down in the Declaration of Helsinki and was approved by the London School of Hygiene & Tropical Medicine Ethics Committee (Reference 28001).

## Variables

**Number of non-communicable diseases (outcome).** We used the responses to the questions asking participants whether they have been diagnosed by a doctor with any of the 14 following NCDs: hypertension, dyslipidemia, diabetes, cancer, lung disease, liver disease, heart disease, stroke, kidney disease, digestive disease, psychiatric problems, memory-related disease, arthritis, and asthma (0 = *no*; 1 = *yes*). We counted the number of concurrent NCDs for each participant in each wave [30].

**Famine exposure (predictor, factorial approach).** We used the responses to the question asking "Between 1958–1962 did you and your family experience starvation?" to build a binary variable differentiating participants who were unexposed to the famine (i.e., who answered "no") from participants who were exposed (i.e., who answered "yes").

**Life stages (predictor, factorial approach).** We adopted a life course perspective and included participants ranging from in-utero to age of 40 at famine exposure. We used age at

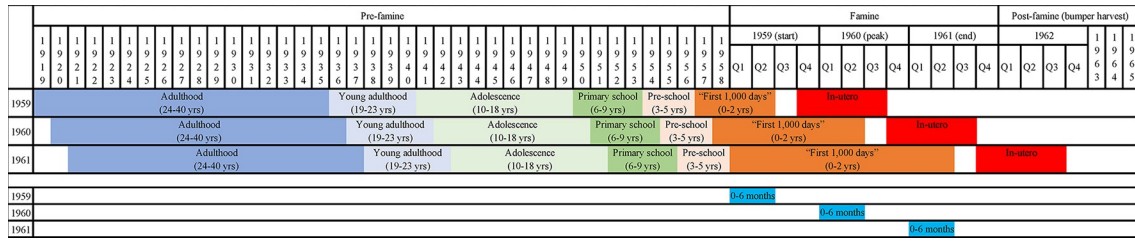

**Fig 1. Categorization scheme of the seven life stages.** *Note.* Columns refer to birth year/quarter, rows refer to reference year of 1959, 1960, and 1961. Q1, Q2, Q3, and Q4 refer to the first, second, third, and fourth quarter of the year. yrs refers to age in years.

famine exposure on the 1 January 1959 to categorize participants into seven life stages adapted from definitions commonly used in major global policies and strategies [31] (**Fig 1**):

(1) **"In-utero"**—those born between 1 October 1959 and 30 September 1960;

(2) **"First 1,000 days" infants** (0–2 years) (reflecting the focus on this age group by the "Scaling Up Nutrition Movement" [32])—those born between 1957 and 30 June 1959, among whom participants born between 1 January 1959 and 30 June 1959 were categorized as "0–6 months";

(3) **"Pre-school" children** (3–5 years)—those born between 1954 and 1956;

(4) **"Primary school" children** (6–9 years)—those born between 1950 and 1953;

(5) **"Adolescents"** (10–18 years)—participants born between 1941 and 1949;

(6) **"Young adults"** (19–23 years)—participants born between 1936 and 1940;

(7) **"Adults"** (24–40 years)—participants born between 1919 and 1935.

**Famine exposure-life stages (predictor, concatenation approach).** We used the variable "famine exposure" (i.e., no/yes) and the variable "life stages" (i.e., in-utero to adulthood) to build a superordinate categorical variable of famine exposure-life stages comparing unexposed participants with exposed participants from in-utero to in adulthood. Specifically, we concatenated the two variables to categorize participants into eight groups: (1) unexposed to the famine during their life course (i.e., control group), (2) exposed in-utero, (3) exposed in the "first 1,000 days" (0–2 years), (4) exposed in pre-school age (3–5 years), (5) exposed in primary school age (6–9 years), (6) exposed in adolescence (10–18 years), (7) exposed in young adulthood (19–23 years), and (8) exposed in adulthood (24–40 years).

**Severity of famine exposure (predictor).** We used the responses to the questions following up the famine exposure question to build a variable differentiating participants who were unexposed to famine from participants who were *moderately* exposed to famine (i.e., who answered "yes" to the question "Between 1958–1962 did you and your family move away from the famine-stricken area?"), and participants who were *severely* exposed to famine (i.e., who answered "yes" to either of the questions "Between 1958–1962 had any of your family starved to death?" or "Between 1958–1962 had your family lost any child?").

**Potential confounders.** We controlled for the following potential confounders in our analysis: age, sex (-0.5 = *male*, +0.5 = *female*), residence in later life (0 = *rural*, 1 = *urban*), current marital status (0 = *not married*, 1 = *married*), current working status (0 = *not working*, 1 = *working*), family financial status in childhood (from 1 = *much better* to 5 = *much worse than families in the same community*), education levels (1 = *less than upper secondary*, 2 = *upper*

*secondary and vocational*, 3 = *tertiary*), household income deciles (1 = *bottom 10%*, 10 = *top 10%*), number of diseases in childhood, and number of diseases in adulthood.

### Analytical strategy

**Poisson growth curve models.** To estimate trajectories of the number of NCDs in the later life course, we built a series of two-level growth curve models in which wave-specific observations ($N$ = 39,337 level-1 units) were nested in participants ($K$ = 11,094 level-2 units). Specifically, we built Poisson growth curve models instead of linear growth curve models [33], because our outcome variable (the number of NCDs) was a count variable following a Poisson distribution. The overdispersion test revealed that the assumption of the equidispersion of the Poisson regression was not violated, $\chi^2$ (2, $N$ = 39,337) = 15,849, $p$ = 1.00.

**Associations between famine exposure, life stage, and later-life NCDs: Two approaches.** We used two approaches that have been used in previous studies to test the associations between famine exposure, life stage, and later-life NCDs: the factorial approach [34–36] and the concatenation approach [22, 37].

*The factorial approach.* In the first approach, we tested whether the dichotomous variable "famine exposure" (unexposed vs. exposed) interacted with the variable "life stages" in predicting the number of NCDs. As life stage at famine exposure is partially dependent on the current age of the participant, we included the age variable in our model to estimate the effect of life stage at famine exposure independent of the current age [38]. We regressed the number of NCDs on three focal predictors: (i) famine exposure, (ii) life stages, (iii) famine exposure × life stages, and (iv) age (see Eq 1), with and without control variables.

$$
\begin{aligned}
\log(\lambda_{ij}) = {} & (\beta_{00} + u_{0j}) + \beta_{01} \times \text{Famine exposure}_j + \sum_{k=1}^{6} (\beta_{0(1+k)} \times \text{Life stage } k_j) \\
& + \sum_{k=1}^{6} (\beta_{0(7+k)} \times \text{Famine exposure}_j \times \text{Life stage } k_j) + \beta_{10} \times \text{Age}_{ij} + \beta_{ij} \\
& \times \text{Control}_{ij}
\end{aligned}
\tag{Eq 1}
$$

The outcome variable follows a Poisson distribution ($Y_{ij} \sim \text{Poisson}(\lambda_{ij})$); $i$ = 1, 2, . . ., $N$ (wave-specific observations), $j$ = 1, 2, . . ., $K$ [participants]; "Life stage" 1 to 6 correspond to in-utero, the "first 1,000 days", pre-school, primary school, adolescence, and young adulthood, respectively; $\beta_{ij} \times \text{Control}_{ij}$ represents a vector of control variables; and $u_{0j}$ represents the participant-level residuals.

Although the factorial approach enabled us to directly test whether the association between famine exposure and later-life NCDs varies across life stages, the fact that there are seven life stages consumes a lot of degrees of freedom (i.e., seven for the main effect, and seven others for the interaction) and represents a potential threat to statistical power. Thus, we additionally used an alternative, more parsimonious and focal approach.

*The concatenation approach.* In the second approach, we tested whether the superordinate categorical variable "famine exposure-life stages" predicted the number of NCDs, comparing unexposed participants with participants exposed in-utero, in the "first 1,000 days" (0–2 years), in pre-school age (3–5 years), in primary school age (6–9 years), adolescence (10–18 years), in young adulthood (19–23 years), and in adulthood (24–40 years). We included the age variable in our model for the same reason mentioned above. We regressed the number of NCDs on two focal predictors: (i) famine exposure-life stages, and (ii) age (see Eq 2), with and without control variables.

$$
\log(\lambda_{ij}) = (\beta_{00} + u_{0j}) + \sum_{k=1}^{7} (\beta_{0k} \times \text{Famine exposure} \frown \text{life stage } k_j) + \beta_{10} \times \text{Age}_{ij} + \beta_{ij} \times \text{Control}_{ij}
\tag{Eq 2}
$$

The outcome variable that follows a Poisson distribution ($Y_{ij} \sim \text{Poisson}(\lambda_{ij})$); $i$ = 1, 2, . . ., $N$ (wave-specific observations); $j$ = 1, 2, . . ., $K$ [participants]; "Famine exposure-life stage" 1 to 7

corresponds to exposed to famine in-utero, in the "first 1,000 days", in pre-school, in primary school, in adolescence, in young adulthood, and in adulthood, respectively; $\beta_{ij} \times \text{Control}_{ij}$ represents a vector of control variables; and $u_{0j}$ represents the participant-level residuals.

**Sensitivity analyses.** We conducted two sets of sensitivity analyses to test the robustness of the associations between famine exposure, life stage, and later-life NCDs. First, to test if these associations were robust across the famine years as the famine evolved, we used two alternative reference dates of 1 January 1960 and 1 January 1961 to categorize participants into the seven life stages mentioned above (**Fig 1**). Second, to test is these associations were robust to the conceptualisation of life stages, we used an alternative scheme of life stages based on Erikson's developmental stages [39] (**S1 Text**).

We ran the Poisson growth curve models using the glmer function from the lme4 package (version 1.1–26) [40] in R (version 4.0.2). The R scripts to reproduce our findings are available via the Open Science Framework (OSF): https://osf.io/6zy43/?view_only=bacb4fa4927d4019a0c298da00082d91.

## Results

Our final sample comprised 39,337 observations from 11,094 older adults (for the flow diagram of study participants, see **Fig 2**). As expected from the scope of the Chinese famine of 1959–1961, the majority of the analytical sample (9,257 participants, 83%) was exposed to famine whereas a small proportion of the analytical sample (1,837 participants, 17%) was unexposed to famine. **Table 1** details the number of participants by sex and famine exposure at each life stage, and **Table 2** describes the sample characteristics.

### Associations between famine exposure, life stage, and later-life NCDs

**Factorial approach.** In our first set of models using the factorial approach (adjusted for confounders), we observed a main effect of famine exposure: compared with unexposed

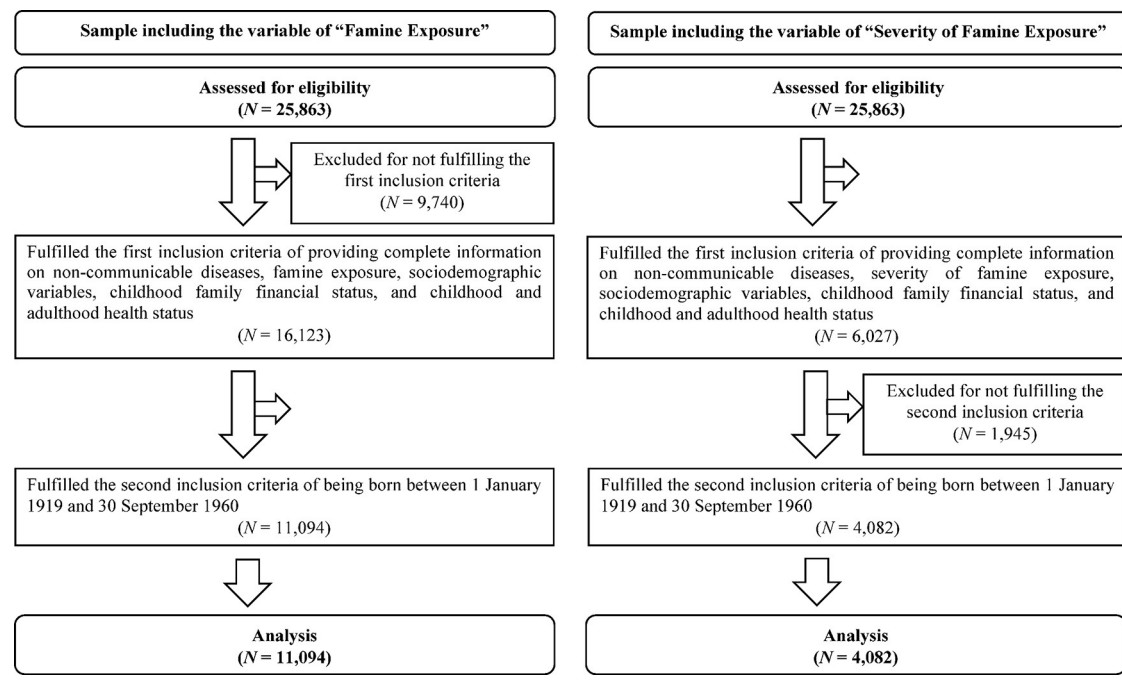

**Fig 2. Flow diagram of study participants.**

**Table 1. Number of study participants, stratified by sex and self-reported famine exposure at each life stage.**

|  | Overall (11,094 participants) | | Male (5,446 participants) | | Female (5,648 participants) | |
|---|---|---|---|---|---|---|
|  | Exposed (9,257) | Unexposed (1,837) | Exposed (4,626) | Unexposed (820) | Exposed (4,631) | Unexposed (1,017) |
| In-utero | 311 | 82 | 144 | 37 | 167 | 45 |
| The "first 1,000 days" (0–2 years)[1] | 914 (133) | 225 (39) | 448 (56) | 102 (21) | 466 (77) | 123 (18) |
| Pre-school (3–5 years) | 1,532 | 319 | 767 | 124 | 765 | 195 |
| Primary school (6–9 years) | 2,046 | 287 | 980 | 123 | 1,066 | 164 |
| Adolescence (10–18 years) | 3,001 | 489 | 1,554 | 217 | 1,447 | 272 |
| Young adulthood (19–23 years) | 813 | 239 | 437 | 116 | 376 | 123 |
| Adulthood (24–40 years) | 640 | 196 | 296 | 101 | 344 | 95 |

[1] Numbers in parentheses refer to the number of participants aged 0–6 months.

participants, participants exposed to famine had a 14% increased risk of NCDs in later life, IRR = 1.14, 95% CI [1.09, 1.19], $p < .001$. However, we did not observe an interaction between famine exposure and life stages: the overall detrimental effect of famine exposure did not vary across life stages when participants were exposed, $\chi^2$ (6, N = 39,337) = 1.38, $p = .241$.

**Concatenation approach.** In our second set of models using the concatenation approach (adjusted for confounders), we observed a main effect of the famine exposure-life stages: compared with participants unexposed to famine over their life course, participants exposed to famine at different life stages were affected differently, $\chi^2$ (7, N = 39,337) = 1117.00, $p < .001$ (Fig 3; for the full results, see Table 3). Specifically, compared with participants unexposed to famine over their life course (control group): (i) participants exposed to famine in-utero and in the "first 1,000 days" of their life had an approximately 90% increased risk of NCDs in later life (for in-utero group, IRR = 1.90, 95% CI [1.70, 2.12], $p < .001$; for 0–2 years group, IRR = 1.86, 95% CI [1.73, 2.00], $p < .001$, among which for 0–6 months group, IRR = 1.95, 95% CI [1.67, 2.29], $p < .001$); (ii) participants exposed to famine in pre-school and primary school (i.e., before age 10) had an approximately 50% increased risk of NCDs in later life (for pre-school group (3–5 years), IRR = 1.56, 95% CI [1.47, 1.66], $p < .001$; for primary school

**Table 2. Sample characteristics, stratified by self-reported famine exposure.**

|  | Overall (11,094 participants) | Exposed (9,257 participants) | Unexposed (1,837 participants) |
|---|---|---|---|
| Females (%) | 50.9 | 50.0 | 55.4 |
| Age (mean; in years) | 65.6 (SD = 7.7) | 65.5 (SD = 7.5) | 66.3 (SD = 8.6) |
| Childhood family financial status (mean; 1 = *much better*, 5 = *much worse than other families*) | 3.55 (SD = 0.99) | 3.62 (SD = 0.98) | 3.20 (SD = 0.96) |
| Equivalized Income (mean; in CNY) | 13,705 (SD = 26,229) | 13,148 (SD = 26,633) | 16,514 (SD = 23,903) |
| Educational levels (%) |  |  |  |
| less than lower secondary | 91.0 | 91.4 | 88.7 |
| upper secondary or vocational | 7.7 | 7.4 | 8.9 |
| tertiary | 1.3 | 1.1 | 2.3 |
| Number of non-communicable diseases (mean) | 2.01 (SD = 1.56) | 2.04 (SD = 1.57) | 1.85 (SD = 1.50) |
| Number of childhood health conditions (mean) | 0.08 (SD = 0.33) | 0.08 (SD = 0.33) | 0.07 (SD = 0.32) |
| Number of adulthood health conditions (mean) | 0.38 (SD = 0.74) | 0.39 (SD = 0.74) | 0.34 (SD = 0.70) |
| Rural residence in later life (%) | 62.3 | 63.1 | 58.0 |
| Currently married (%) | 84.9 | 85.4 | 82.3 |
| Currently working (%) | 57.8 | 59.1 | 51.4 |

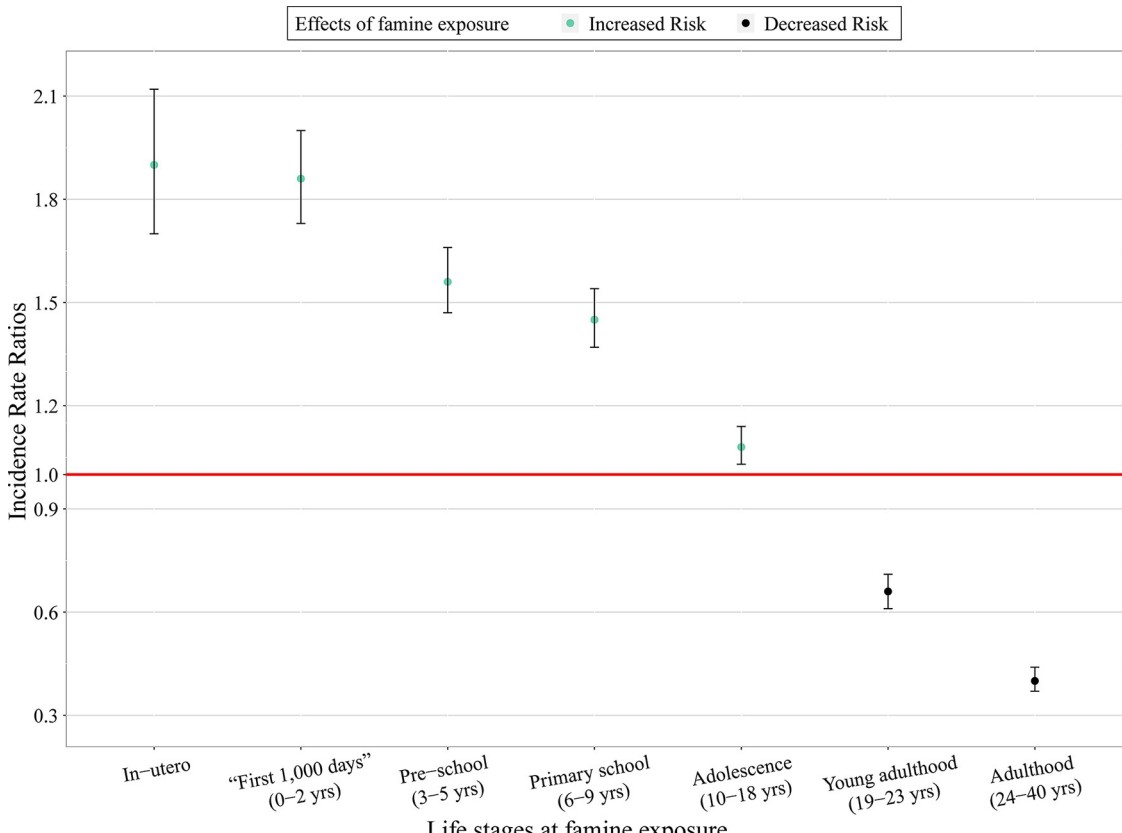

**Fig 3. Associations between exposure to the 1959–1961 Chinese famine, life stage, and NCDs in later life (the concatenation approach).**

**Table 3. Associations between exposure to the 1959–1961 Chinese famine, life stage, and NCDs in later life (the concatenation approach).**

|  | IRRs | 95% CI |
|---|---|---|
| Unexposed (control group) | 1.00 | Reference |
| Exposed in-utero | 1.90*** | 1.70–2.12 |
| Exposed in the "first 1,000 days" (0–2 years)[1] | 1.86*** (1.95***) | 1.73–2.00 (1.67–2.29) |
| Exposed in pre-school age (3–5 years) | 1.56*** | 1.47–1.66 |
| Exposed in primary school age (6–9 years) | 1.45*** | 1.37–1.54 |
| Exposed in adolescence (10–18 years) | 1.08** | 1.03–1.14 |
| Exposed in young adulthood (19–23 years) | 0.66*** | 0.61–0.71 |
| Exposed in adulthood (24–40 years) | 0.40*** | 0.37–0.44 |

*Note*. IRRs = Incidence Rate Ratios.

*$p < .05$

**$p < .01$

***$p < .001$

Adjusted for age, sex, later-life residence, marital status, current working status, childhood family financial status, education, household income, number of diseases in childhood, and number of diseases in adulthood.

[1] Numbers in parentheses refer to the IRRs and 95% CI for participants exposed in the 0–6 months.

group (6–9 years), IRR = 1.45, 95% CI [1.37, 1.54], $p < .001$); (iii) participants exposed to famine in adolescence (10–18 years) had a 8% increased risk of NCDs in later life, IRR = 1.08, 95% CI [1.03, 1.14], $p < .01$; whereas (iv) participants exposed to famine after age 18 had a decreased risk of NCDs in later life (for young adulthood group (19–23 years), IRR = 0.66, 95% CI [0.61, 0.71], $p < .001$; for adulthood group (24–40 years), IRR = 0.40, 95% CI [0.37, 0.44], $p < .001$).

**Sensitivity analyses.** We conducted two sets of sensitivity analyses to test the robustness of the associations between famine exposure, life stage, and later-life health. The results observed above were confirmed in the sensitivity analyses that used two alternative reference dates (**S1 Table**) and an alternative scheme of life stages based on Erikson's developmental stages [39] (**S2 Table**).

## Associations between severity of famine exposure and later-life NCDs

We ran the same type of Poisson growth curve model as above, but this time we used severity of famine exposure (unexposed, moderately exposed, and severely exposed) as the focal predictor. Compared with unexposed participants, participants moderately or severely exposed to the famine had an increased risk of NCDs in later life, IRR = 1.18, 95% CI [1.09, 1.28], $p < .001$ and IRR = 1.24, 95% CI [1.17, 1.32], $p < .001$, respectively. However, the difference in the risk of NCDs in later life between moderately exposed participants and those severely exposed to famine was not statistically significant, $\beta = 0.05$, 95% CI [-0.05, 0.15], $p = .454$ (for the full results, see **S3 Table**).

## Sex-specific associations between famine exposure and later-life NCDs

We ran the same type of Poisson growth curve model as above and included the interaction between famine exposure and sex. The overall detrimental famine effects did not differ between females and males, IRR = 0.98, 95% CI [0.90, 1.07], $p = .703$ (for the full results, see **S4 Table**).

## Discussion

In this study, we adopted a life course perspective and analyzed whether exposure to the 1959–1961 Chinese famine was associated with NCDs in later life. We addressed the problem of information bias by using a more direct measure of famine exposure and the severity of famine exposure. We also extended existing studies by testing whether this association depends on the life stage at exposure, the severity of famine exposure, and sex. We observed three main findings:

First, consistent with the few existing studies [22, 37], our analysis using the concatenation approach revealed that individuals exposed to famine at different life stages were affected differently—and the famine effects were independent of age/cohort differences (see **S5 Table**). Compared with individuals unexposed to famine over their life course, individuals exposed to famine before age 18 had a higher risk of NCDs in later life. The risk was particularly high for those exposed to famine in-utero and in the "first 1,000 days" of life, namely, between 0–2 years (particularly 0–6 months). The long-term detrimental health consequences of famine exposure before age 18 can be explained by the wide time window of development and growth [41]. Development during the "first 1,000 days" is particularly rapid and critical, and individuals exposed during this period are most affected by adverse exposures. One widely acknowledged mechanism is the capacity-load model [9]. Poor nutrition and slow/rapid growth during the critical developmental period constrain capacity for homeostasis and elevate metabolic load which increases the risk for NCDs in the long term.

In contrast, compared with those unexposed to famine over their life course, individuals exposed to famine in young adulthood (19–23 years) or adulthood (24–40 years) had a decreased risk of NCDs in later life. This may be explained by the combined effect of food allocation and mortality selection (survivor bias): during the famine adults prioritized food allocation to infants and children in the family [42]. As a result, only the fittest adults survive the famine, which may explain why these adult survivors of famine are generally healthier than their unexposed counterparts.

Second, also consistent with the few existing studies [35], we found that although individuals moderately or severely exposed to the famine had a higher risk of NCDs in later life than unexposed individuals, the difference in the risk of NCDs between moderately and severely exposed individuals was not statistically significant. Future studies are warranted to investigate the possible dose-response effects of famine exposure severity on later-life health. We recognize that we had only retrospective and imperfect measures of famine exposure severity, and famine exposure severity might not be fully captured by the available data (e.g., we had no measure of an individual's nutritional/anthropometric status at the time of famine). Hence the true effects may have been underestimated. It is also possible that "healthy survivor" bias again applies and that those most affected were most likely to die in the short or medium term (see **S6 Table**).

Third, in contrast to some previous studies that reported a more pronounced detrimental impact of famine exposure among females than males [43, 44], we found that the association between famine exposure and NCDs in later life did not differ between females and males. The more pronounced detrimental impact of famine exposure among females than males reported by some previous studies may be biased by the healthy survivor effect among males [24] and the food allocation priority given to sons in the family [25]. Future studies are warranted to investigate the possible sex-specific mechanisms underlying the famine exposure and later-life health [45]. Many mechanisms are possible, but which are active and how they vary in different settings is poorly understood.

We acknowledge four main limitations. First, the retrospective measures of famine exposure and severity of famine exposure were self-reported and might be subject to recall bias and misclassification. However, a recent review points out that the information on major life events collected by interview-based procedures has acceptable levels of reliability [46]. Second, we could not control for birth weight or gestational age, which might be important confounders, because information on these is unavailable. However, several studies showed that the association between famine exposure and NCDs remained after controlling for birth weight [47]. Third, life stages were categorized based on the age at exposure. Future studies should consider the context and transitions into adulthood when categorizing life stages (e.g., experiencing famine during transitions to entering the labor market, starting a family, or having a child). Last, survivor bias might be part of the story. Future studies need to evaluate the effects of survivor bias by investigating the possible selection mechanisms and adjust the effect estimates of famine exposure [48].

## Implications for policy and practice

Our findings have several implications for policy and practice in global health. First, our finding that famine exposure in childhood and adolescence is associated with later-life NCDs is an important argument for seeing undernutrition/famine prevention as a long-term investment rather than a short-term cost. Second, the particularly marked long-term effects if famine is experienced in-utero and in the "first 1,000 days" further justify the current focus of nutrition programmes on this period [32]. Third, our observation that infants aged 0–6 months are also

especially vulnerable justifies WHO's current focus on growth failure in this age group in upcoming wasting guidelines [49, 50]. Finally, the fact that older children and adolescents also experience long-term effects should remind policy-makers and practitioners that these groups also matter and that life course approaches are essential to future health and wellbeing.

## Conclusion

Our research analyzed a large-scale, nationally representative, longitudinal study, which provides both prospective data and retrospective data: the China Health and Retirement Longitudinal Study (CHARLS, approximately 26,000 participants, 2011–2018, aged 45 and above). Our findings show that famine exposure has long-term detrimental consequences for later-life NCDs and affects individuals exposed at different life stage ranging from in-utero until young adulthood. These findings suggest that in an individual's life course, in-utero and the "first 1,000 days" is a particularly sensitive time window of development and growth but that this time window remains open until young adulthood. This represents a powerful argument for actions to protect nutrition throughout this period: such actions should be viewed as an investment, with beneficial short-term as well as long-term health and socioeconomic consequences, not only for individual children and families, but also for nations and the world.

## Supporting information

**S1 Text. Categorization of life stages based on Erikson's developmental stages.**
(DOCX)

**S1 Table. Associations between exposure to the 1959–1961 Chinese famine, life stage, and later-life NCDs (based on life stages commonly used in major global policies and strategies).**
(DOCX)

**S2 Table. Associations between exposure to the 1959–1961 Chinese famine, life stage, and later-life NCDs (based on Erikson's developmental stages).**
(DOCX)

**S3 Table. Associations between severity of exposure to the 1959–1961 Chinese famine and later-life NCDs.**
(DOCX)

**S4 Table. Sex-specific associations between exposure to the 1959–1961 Chinese famine and later-life NCDs.**
(DOCX)

**S5 Table. Age of NCDs onset among participants exposed to famine at different life stage.**
(DOCX)

**S6 Table. Associations between severity of the 1959–1961 Chinese famine and later-life NCDs.**
(DOCX)

## Acknowledgments

The authors would like to thank the chairs, panelists, discussants, and audience at the Society for Longitudinal and Lifecourse Studies (SLLS) 2022 Annual Conference and the Gerontological Society of America (GSA) 2022 Annual Scientific Meeting for their feedback in developing

this study. This study uses data from the Harmonized CHARLS dataset and codebook (Version D, 2011–2018, Waves 1, 2, 3 and 4) developed by the Gateway to Global Aging Data, as well as the CHARLS Life History data (2014) developed by the China Center for Economic Research, Institute of Social Science Survey at Peking University. For more information, please refer to https://g2aging.org/.

## Author Contributions

**Conceptualization:** Mengling Cheng, Nicolas Sommet, Dario Spini.

**Data curation:** Mengling Cheng.

**Formal analysis:** Mengling Cheng.

**Methodology:** Mengling Cheng, Nicolas Sommet.

**Software:** Mengling Cheng.

**Visualization:** Mengling Cheng.

**Writing – original draft:** Mengling Cheng.

**Writing – review & editing:** Mengling Cheng, Nicolas Sommet, Marko Kerac, Daniela S. Jopp, Dario Spini.

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
