## [Decision Letter · Decision Letter 0]

2 May 2023

PGPH-D-23-00113

Exposure to the 1959-1961 Chinese famine and risk of non-communicable diseases in later life: a life course perspective

Dear Dr. Cheng,

Thank you for submitting your manuscript to PLOS Global Public Health. After careful consideration, we feel that it has merit but does not fully meet PLOS Global Public Health’s publication criteria as it currently stands. Therefore, we invite you to submit a revised version of the manuscript that addresses the points raised during the review process.

We look forward to receiving your revised manuscript.

Kind regards,

Rajesh Sharma, Ph.D.

Academic Editor

Journal Requirements:

2. Please send a completed 'Competing Interests' statement, including any COIs declared by your co-authors. If you have no competing interests to declare, please state "The authors have declared that no competing interests exist". Otherwise please declare all competing interests beginning with the statement "I have read the journal's policy and the authors of this manuscript have the following competing interests:"

3. Please amend your detailed Financial Disclosure statement. This is published with the article. It must therefore be completed in full sentences and contain the exact wording you wish to be published.

a. State what role the funders took in the study. If the funders had no role in your study, please state: “The funders had no role in study design, data collection and analysis, decision to publish, or preparation of the manuscript.”

b. If any authors received a salary from any of your funders, please state which authors and which funders.

4. Please provide separate figure files in .tif or .eps format only and remove any figures embedded in your manuscript file. Please also ensure all files are under our size limit of 10MB.

5. We notice that your supplementary file are included in the manuscript file. Please remove them and upload them with the file type 'Supporting Information'. Please ensure that each Supporting Information file has a legend listed in the manuscript after the references list.

Additional Editor Comments (if provided):

Minor Revision

Reviewers' comments:

Reviewer's Responses to Questions

**Comments to the Author**

1. Does this manuscript meet PLOS Global Public Health’s publication criteria? Is the manuscript technically sound, and do the data support the conclusions? The manuscript must describe methodologically and ethically rigorous research with conclusions that are appropriately drawn based on the data presented.

Reviewer #1: Yes

Reviewer #2: Yes

2. Has the statistical analysis been performed appropriately and rigorously?

Reviewer #1: Yes

Reviewer #2: Yes

3. Have the authors made all data underlying the findings in their manuscript fully available (please refer to the Data Availability Statement at the start of the manuscript PDF file)?

Reviewer #1: Yes

Reviewer #2: Yes

4. Is the manuscript presented in an intelligible fashion and written in standard English?

Reviewer #1: Yes

Reviewer #2: Yes

5. Review Comments to the Author

Reviewer #1: Generally, the paper is supremely written with clearly explained methods, and statistical approaches seem to fit the objectives and the identified gaps. In addition, the study hypothesis and key messages are presented clearly and concisely. The relevance of the study to research, practice, and policy is well emphasized in the key messages section.

The introduction is well written, but the sentence structure could be improved for clarity. For example, the sentence "To add to the literature and overcome some of these limitations, we adopted a life course perspective and used more robust methods to investigate the association between exposure to the 1959-1961 Chinese famine and later-life NCDs and if this association depends on: life stage (age) at exposure, the severity of exposure, and sex" could be split into two sentences for better readability.

The 4th sentence of the first paragraph of the introduction seems to introduce a different concept from the paragraph's main idea. Making a separate paragraph would help

The last sentence of the second paragraph starts with "these are" Since more than one aspect is described in the immediate context, it would be better if "these are..." Is precisely defined

Under the "Overview of the Study", there is a mention of "...to add to the literature and overcome some of the limitations ". Please, precisely clary these limitations and how they were overcome. Consequently, show the limitations you failed to overcome and suggest how they should be overcome. Moreover, this failure to overcome these other limitations should be reflected in the conclusions and recommendations

Under the same

"Technically, avoid writing objectives in the form of questions. Please, change them into statements" The first sentence of the conclusion should be revised. It should reflect the data collection approach used

The following sentences had some grammatical errors, and below are suggested corrections:

• Existing literature suggests that undernutrition in childhood has immediate detrimental and long-term adverse health consequences, including increasing the risk of NCDs in later life.

• Life course epidemiology suggests that the timing of exposure is critical and that the effect of exposure during a specific period may differ from ……..

• Thus, individuals exposed to famine at different life stages may be affected differently.

• A major source of information bias is the misclassification of famine exposure …….

• We measured famine exposure and severity of famine exposure using the individual self-reported famine experience, life stages using the age at famine exposure, and later-life health using the number of NCDs. We built a series of Poisson growth curve models to investigate the association between famine exposure and later-life NCDs.

• Regarding the severity of famine exposure (predictor), we used the responses to the questions following up the famine exposure question to build a variable differentiating participants who were unexposed to famine Sensitivity analysis

• We conducted two sets of sensitivity analyses to test the robustness of the associations between famine exposure, life stage, and later life health.

• The results observed above were in the sensitivity analysis that used two alternative reference dates (Table S1) and an alternative scheme of life stages based on Erikson’s38 developmental stages (Table S2).

• However, the difference in the risk of NCDs in later life between moderately exposed participants and those severely exposed to famine failed.

• We addressed the problem of information bias by using a more direct measure of famine exposure and severity of famine exposure. We also extended existing studies by testing whether this association depends on the life stage at exposure, the severity of famine exposure, and sex. We observed three main findings:

• Second, we could not control for birth weight or gestational age, which might be important confounders because information on these is unavailable.

• However, several studies showed that the association between famine exposure and NCDs remains after controlling for birth weight.

• These findings suggest that an individual's life course, in-utero and the “first 1,000 days,” is a particularly sensitive time window of development and growth but that this time window remains open until young adulthood.

Lastly, the correct placement of punctuation mark

Reviewer #2: This is an excellent study. Very concise yet informative. The authors left no room for redundancy. There is the logical flow of ideas. The introduction is well written. The gap and justification for this study was also well stated. The authors identified limitations and methodological errors/biases in previous studies which is very critical for the understanding and readers appreciation of this current study. I love the fact that, the authors identified all the potential confounding factors right from the beginning of the study and controlled for them. I have just two concerns:

1. The study design used was not clear or mentioned in the methods section until at the tail end of the manuscript (during their discussion of study limitations) which leaves readers wondering for long.

2. the sampling method and technique used in the selection of the sample population. I see a flow diagram describing reasons for excluding certain participants and the final sample size used. but that still does not clearly answer the question on the sampling method used. this should have been briefly discussed in the methods section.

Good luck and congratulations for a good work well done. Thank you for the extra efforts put into this work.

6. PLOS authors have the option to publish the peer review history of their article (what does this mean?). If published, this will include your full peer review and any attached files.

**Do you want your identity to be public for this peer review?** For information about this choice, including consent withdrawal, please see our Privacy Policy.

Reviewer #1: **Yes: **Patricia Kamanga

Reviewer #2: No

---

## [Decision Letter · Decision Letter 1]

21 Jun 2023

Exposure to the 1959-1961 Chinese famine and risk of non-communicable diseases in later life: A life course perspective

PGPH-D-23-00113R1

Dear M.A. Cheng,

We are pleased to inform you that your manuscript 'Exposure to the 1959-1961 Chinese famine and risk of non-communicable diseases in later life: A life course perspective' has been provisionally accepted for publication in PLOS Global Public Health.

Best regards,

Rajesh Sharma, Ph.D.

Academic Editor

Accept

Reviewer Comments (if any, and for reference):

Reviewer's Responses to Questions

**Comments to the Author**

1. If the authors have adequately addressed your comments raised in a previous round of review and you feel that this manuscript is now acceptable for publication, you may indicate that here to bypass the “Comments to the Author” section, enter your conflict of interest statement in the “Confidential to Editor” section, and submit your "Accept" recommendation.

Reviewer #1: All comments have been addressed

Reviewer #2: All comments have been addressed

2. Does this manuscript meet PLOS Global Public Health’s publication criteria? Is the manuscript technically sound, and do the data support the conclusions? The manuscript must describe methodologically and ethically rigorous research with conclusions that are appropriately drawn based on the data presented.

Reviewer #1: Yes

Reviewer #2: Yes

3. Has the statistical analysis been performed appropriately and rigorously?

Reviewer #1: Yes

Reviewer #2: Yes

4. Have the authors made all data underlying the findings in their manuscript fully available (please refer to the Data Availability Statement at the start of the manuscript PDF file)?

Reviewer #1: Yes

Reviewer #2: Yes

5. Is the manuscript presented in an intelligible fashion and written in standard English?

Reviewer #1: Yes

Reviewer #2: Yes

6. Review Comments to the Author

Reviewer #1: The article is exceptionally well written, demonstrating a profound understanding of the subject matter and presenting a comprehensive analysis. The methodology employed is rigorous, the data analysis and interpretation are thorough, with clear and concise descriptions of statistical methods used. The findings are presented in a logical manner, effectively supporting the article's main arguments. I believe that publishing it in this journal would greatly contribute to the advancement of knowledge in the field.

Reviewer #2: (No Response)

7. PLOS authors have the option to publish the peer review history of their article (what does this mean?). If published, this will include your full peer review and any attached files.

**Do you want your identity to be public for this peer review?** For information about this choice, including consent withdrawal, please see our Privacy Policy.

Reviewer #1: **Yes: **Patricia Kamanga

Reviewer #2: No
